# Tissue Response to a Porous Collagen Matrix Used for Soft Tissue Augmentation

**DOI:** 10.3390/ma12223721

**Published:** 2019-11-11

**Authors:** Jordi Caballé-Serrano, Sophia Zhang, Luca Ferrantino, Massimo Simion, Vivianne Chappuis, Dieter D. Bosshardt

**Affiliations:** 1Robert K. Schenk Laboratory of Oral Histology, School of Dental Medicine, University of Bern, 3010 Bern, Switzerland; jordicase@uic.es (J.C.-S.); sophia.zhang@zmk.unibe.ch (S.Z.); 2Department of Oral and Maxillofacial Surgery, School of Dental Medicine, Universitat Internacional de Catalunya, 08190 Barcelona, Spain; 3Department of Oral Surgery and Stomatology, School of Dental Medicine, University of Bern, 3010 Bern, Switzerland; vivianne.chappuis@zmk.unibe.ch; 4Department of Aesthetic Dentistry, Istituto Stomatologico Italiano, University of Milan, 20122 Milan, Italy; luca.ferrantino@gmail.com; 5UOC Chirurgia Maxillofacciale e Odontostomatologia, Università di Milano, Fondazione IRCCS Cà Granda, 20122 Ospedale, Italy; msimion@studiosimion.it

**Keywords:** VCMX, Fibro-Gide, collagen matrix, immunohistochemistry, GTR, GBR, volume-stable collagen matrix

## Abstract

A short inflammatory phase and fast ingrowth of blood vessels and mesenchymal cells are essential for tissue integration of a biomaterial. Macrophages play a key role in this process. We investigated invasion of macrophages, blood vessels, and proliferating cells into a highly porous and volume-stable collagen matrix (VCMX) used for soft tissue augmentation around teeth and dental implants. The biomaterial was implanted in submucosal pouches in the canine maxilla, and the tissue response was analyzed at six different time points. Immunohistochemistry was done for proliferating cells (PCNA), macrophages (MAC387), multinucleated giant cells (CD86), and blood vessels (TGM2). Blood rapidly filled the VCMX pores. During the first week, MAC387+ cells populated the VCMX pores, blood vessels and PCNA+ cells invaded the VCMX, and CD86+ scattered cells were observed. At 15 days, MAC387+ cells were scanty, blood vessels had completely invaded the VCMX, the number of proliferating cells peaked, and fibroblasts appeared. At 30 days, MAC387+ were absent, the numbers of proliferating and CD86+ cells had declined, while blood vessel and fibroblast numbers were high. At 90 days, residual VCMX was well-integrated in soft connective tissue. In conclusion, the VCMX elicited a short inflammatory phase followed by rapid tissue integration.

## 1. Introduction

Biomaterials are widely used in the dental field to overcome the drawbacks of autogenous tissue transplantation. Lack of an appropriate periodontal or peri-implant soft tissue width and thickness can compromise esthetics, function, or survival of teeth and dental implants [1,2]. Increase of soft tissue thickness is a proven strategy to increase keratinized tissues [3,4], quality of mucosa around teeth [1] and dental implants [5,6], as well as aesthetics [7]. Autogenous connective tissue grafts are used to correct oral tissue deficiencies [8] and still remain the gold standard for soft tissue augmentation due to the good maintenance of volume and lack of adverse effects. However, autogenous grafts always require a donor site, causing morbidity and possible damage of neighboring anatomical structures [9]. Therefore, biomaterials have been developed as alternatives to increase the amount of soft tissue and keratinization [10]. They include allografts, xenografts, and synthetic soft tissue substitutes [11,12,13]. Among the most widely used biomaterials for soft tissue augmentation are collagen matrices of porcine origin. Although the latter have demonstrated optimal clinical outcomes with uneventful tissue integration [10], the volume gain tends to be lower when compared to the subepithelial connective tissue graft [14].

Recently, a new, highly porous, and volume-stable collagen-based matrix (VCMX) was developed for soft tissue augmentation around teeth and dental implants [15]. Such a biomaterial must be biocompatible, allow ingrowth of blood vessels and progenitor cells, and withstand mechanical forces generated as a result of suturing, wound contraction, and mastication, thereby maintaining tissue volume. In vivo and in vitro studies demonstrated optimal mechanical, biological, and anatomical properties of the VCMX [16,17,18,19]. The collagen matrix consists of 60–96% (w/w) porcine collagen type I and III and 4–40% (w/w) elastin, has an average pore diameter of 92 µm, and 93% volume porosity with interconnected pores. Scaffold stiffness was achieved by chemical cross-linking. The scaffold remained elastic even after mechanical forces were applied for 14 days, as evaluated in a dynamic bioreactor test system mimicking the mechanical stresses of the human in vivo situation [18]. Clinically, soft tissue augmentation using VCMX resulted in a volume increase non-inferior to connective tissue grafts at implant sites in the aesthetic zone [20] and a minimal loss of soft tissue thickness six months after implantation [21]. The underlying biological process leading to these positive clinical results is, however, largely unknown. The outcome of wound healing largely depends on the biomaterials’ characteristics (e.g., chemistry, architecture, surface topography). Ideally, a biomaterial should trigger a short and mild inflammatory period followed by a regenerative phase but no chronic foreign body reaction leading to biomaterial encapsulation or even rejection. Macrophages are among the first cells to come in contact with an implanted biomaterial. The biomaterials’ properties can modulate the phenotype of macrophages, as demonstrated for surface characteristics [22,23], and, as a consequence, the shift from inflammation (M1 macrophages) to tissue repair and regeneration (M2 macrophages) may be impaired or delayed. Moreover, the biomaterial may trigger the formation of multinucleated giant cells, which assist in biomaterial degradation or may indicate a foreign body reaction [24,25,26]. Tissue integration is directly related to the inflammatory response and cell invasion into the biomaterial [27]. A short inflammatory phase will ensure an early start of the proliferative phase with mesenchymal cells invading the biomaterial [19]. Proliferation of cells into the biomaterial is crucial for the integration of the biomaterial in the host tissue, denoting a high biocompatibility [28,29]. Furthermore, the rigidity of a biomaterial determines cell invasion and the cell type into which mesenchymal cells will differentiate [30]. Cell invasion into collagen scaffolds depends on the presence of highly interconnected pores [31,32]. Furthermore, cell proliferation during the early phase is dependent on the presence of vascular structures [28]. Endothelial and fibroblastic cells work together, eventually leading to filling the voids of the biomaterial with collagen as part of the extracellular matrix [19].

A first idea about the tissue response and the behavior of the VCMX after implantation was obtained in a recently published, merely descriptive, study [19]. Currently, however, there are no data available on the characterization of cells invading the VCMX and the dynamic changes over time. Therefore, the aim of the present preclinical study is to qualitatively and quantitatively evaluate the inflammatory response and invasion of blood vessels and proliferating cells into this collagenous biomaterial. To this end, the VCMX was implanted in a clinically relevant canine model and immunohistochemistry was used to characterize monocytes/macrophages (MAC387), multinucleated giant cells (CD86), proliferating cells (PCNA), and blood vessels (TGII) invading the VCMX for an observation period of 2 hours to 90 days.

## 2. Materials and Methods 

This preclinical study protocol has been approved by the Ethical Committee of NAMSA (ID number: 0041573) and conducted in accordance with the policies and principles of the Organization for Economic Cooperation and Development Good Laboratory Practice regulations and the European Union Guidelines (86/609/EEC).

### 2.1. Surgical Phase

Detailed protocol of animal selection and surgical phase can be found in a recent publication using the same samples [19]. Briefly, six mature dogs weighing between 11.6 and 14.5 kg were selected for the study. Six maxillary premolars were atraumatically extracted to create two edentulous spaces in each dog. After 90 days, a full thickness flap was elevated, and either a VCMX (Fibro-Gide^®^ prototype, Geistlich Pharma AG, Wolhusen, Switzerland), in contact with the bone, was placed, or the flap was repositioned without the use of a biomaterial (sham group). Wounds were closed by primary intention. Sacrifice of animals was performed at days 0, 4, 7, 15, 30, and 90.

### 2.2. Scanning Electron Microscopy

The VCMX was gold-coated and observed at high-vacuum in a Zeiss 940 DSM Scanning Electron Microscope (Carl Zeiss, Oberkochen, Germany).

### 2.3. Histologic Processing and Descriptive Analysis

Upon receiving the samples, tissue blocks were trimmed and left in 4% formalin for a period of 10 days. Demineralization was performed by leaving the tissue blocks in 10% ethylenediaminetetraacetic acid (EDTA) for 5 weeks. Tissue blocks were divided into 3 sections and embedded in paraffin, methylmethacrylate (MMA), or in acrylic resin (LR White, Sigma–Aldrich, St. Louis, MO, USA). Paraffin sections were cut at 5 μm, while MMA-embedded tissues were cut with a diamond-coated disc and ground to a thickness of 100 μm. LR White sections were cut at 1 μm with a diamond knife on a Reichert Ultracut E microtome (Leica Microsystems, Wetzlar, Germany). MMA-embedded sections were stained with basic fuchsin and toluidine blue, and LR White-embedded sections were double stained with toluidine blue and basic fuchsin. Paraffin embedded sections were stained with hematoxylin/eosin, Giemsa, or Masson’s trichrome. Digital photography was performed using a digital camera (AxioCam MRc, Carl Zeiss, Oberkochen, Germany) connected to a microscope (Axio Imager M2, Carl Zeiss, Oberkochen, Germany).

### 2.4. Preparation for MAC387 Immunohistochemistry 

To visualize cells that belong to the inflammatory linage derived from hematopoietic stem cells such as granulocytes, monocytes, and macrophages, the MAC387 antibody was used. Tissue sections were deparaffinized and rehydrated, washed, and blocked 30 min with defatted milk. Dako EnVisionTM + Dual Link System-HRP (DAB+) (Agilent, Santa Clara, (CA), USA) was used, adding an incubation with Dako Proteinase K during 5 min. The MAC387 antibody (Anti-S100A9+Calprotectin, Abcam, Cambridge, UK) was diluted 1:2000 and used overnight at 4 °C. Counterstaining of the samples was performed with Mayer’s hematoxylin solution (Merck, Darmstadt, Germany).

### 2.5. Preparation for PCNA Immunohistochemistry

To visualize cells that were in a proliferative phase, paraffin-embedded sections underwent inmunohistochemical staining with anti-proliferating cell nuclear antigen (PCNA, clone PC10) antibody (DAKO, Glostrup, Denmark). Tissue sections were deparaffinized and rehydrated. Heat-induced epitope retrieval was performed at 85 °C for 10 min with a citrate solution. Samples were blocked 30 min with defatted milk, and Dako EnVisionTM + Dual Link System-HRP (DAB+) was used. Incubation with the antibody, diluted 1:50, was performed at room temperature for 1 h. Counterstaining of the samples was performed with Mayer’s hematoxylin solution (Merck, Darmstadt, Germany).

### 2.6. Histologic Quantitative Analysis

Quantitative analysis was performed for cells labeled for PCNA and MAC387. Three pictures were taken randomly at the margin and the center of the VCMX through the different time points (4, 7, 15, 30, 90 days). An area of 0.36 mm^2^ was delimited in every picture and positive cells were counted. The paired t-test was used, and the statistical significance was established at *p* < 0.05.

### 2.7. Preparation for CD86 Immunohistochemistry 

Paraffin-embedded sections underwent inmunohistochemical staining with an anti-CD86 antibody (clone EP1158Y, CD86, Abcam). Heat-induced epitope retrieval was performed at 85 °C for 10 min with a citrate solution. Samples were blocked 30 min with defatted milk, and Dako EnVisionTM + Dual Link System-HRP (DAB+) was used. Incubation with the antibody was performed at room temperature for 2 h. The antibody was diluted 1:100. Counterstaining of the samples was performed with Mayer’s hematoxylin solution (Merck, Darmstadt, Germany).

### 2.8. Preparation for TGM2 Immunohistochemistry 

To stain blood vessels, paraffin-embedded sections underwent inmunohistochemical staining with anti-Trans Glutaminase II antibody (clone CUB7402, TGM2, Thermo Fisher Scientific, Waltham, MA, USA). Deparaffinized sections were blocked utilizing 3% hydrogen peroxide, and heat-induced epitope retrieval was performed at 92 °C for 12 min with a citrate solution. Incubation with the 1:100 diluted antibody was performed at room temperature for 1 h. Wash buffer and secondary antibody were purchased from Zytomed Systems GmbH (Berlin, Germany). Counterstaining of the samples was performed with Mayer’s hematoxylin solution (Merck, Darmstadt, Germany).

## 3. Results

No surgical or healing complications were recorded in any of the animals. All tissue samples could be harvested successfully and were processed histologically. The paraffin histology showed the presence of the VCMX close to the bone surface along the maxillary alveolar process at all healing periods (Figure 1A). Because of progressing integration in host tissue, the distinction between biomaterial and surrounding tissue was most readily possible for healing periods up to 30 days. The biomaterial displayed a trabecular structure forming large honeycomb-like interconnected pores (Figure 1B,C) and consisted of an amorphous, sheet-like matrix with embedded, rod-like structures (Figure 1B). In the 90-day sample, residual VCMX was still visible and well-integrated in soft connective tissue. For reasons of consistency and standardization, the following description of the inmunohistochemical results will be confined to the region facing the bone surface (see right rectangle in Figure 1A).

The VCMX was clearly identifiable and demarcated from the surrounding tissues up to 30 days. At 4 days (Figure 2A), the connective tissue surrounding the VCMX showed the typical feature of granulation tissue, i.e., presence of a fibrin network, many erythrocytes, small blood vessels, and some leukocytes. Initial invasion of blood vessels, leukocytes, and mesenchymal cells into the VCMX pores was observed at 4 days and restricted to the border region only. At 7 days (Figure 2B), larger blood vessels and many mesenchymal cells were seen surrounding and invading the collagen scaffold. At 15 days (Figure 2C), still many blood vessels, few inflammatory cells, and many spindle-shaped fibroblasts oriented parallel to newly formed collagen fibers were seen external to the biomaterial. Some residual biomaterial became integrated in the surrounding soft connective tissue. At 30 days (Figure 2D), the surrounding host collagen matrix appeared more mature and the epithelial layer was closed and well-keratinized. At the border, the biomaterial was well-integrated in the surrounding more mature soft connective tissue, and inflammatory cells were scarce. At 90 days (Figure 2E), residual islands of biomaterial were dispersed and well-integrated in the surrounding host soft connective tissue.

### 3.1. Masson’s Trichrome Stain

Immediately after implantation, the pores of the VCMX were mainly filled with erythrocytes and blood plasma. At 4 days, a fibrin network and erythrocytes were present within most pores in the peripheral part of the VCMX, whereas small blood vessels and mesenchymal-like cells were seen in the soft connective tissue adjacent to and invading the VCMX (Figure 2A). At 7 days, blood vessels progressively invaded the VCMX and increased in size and number at the periphery, and the number of scattered invading mesenchymal-like cells increased as well (Figure 2B). At 15 days, the VCMX pores were completely invaded by small blood vessels, mesenchymal-like cells within the pores had increased in numbers, and an extracellular matrix began to fill the pores (Figure 2C). At 30 days, many blood vessels, predominantly mesenchymal cells now resembling slender fibroblasts, and an extracellular matrix filled the VCMX pores (Figure 2D). At 90 days, packages of residual VCMX were fully integrated in soft connective tissue and the number of blood vessels was reduced and confined to the soft connective tissue surrounding the residual VCMX packages (Figure 2E,K). Compared to the marginal VCMX, the central region showed a delay in coagulation, invasion of both blood vessels and mesenchymal-like cells, and formation of extracellular matrix within the VCMX pores (Figure 2F–I,K). This extracellular matrix was fibrous and increasingly filled the pores of the VCMX starting from the periphery at 7 days and progressing towards the center of the VCMX.

### 3.2. Immunohistochemistry with the Anti-MAC387 Antibody

MAC387^+^ cells were seen in the soft connective tissue next to the VCMX and in the peripheral part of the VCMX at 4 (Figure 3A) and 7 (Figure 3B) days. In the peripheral part of the VCMX, the number of positive cells was highest at 4 days, whereas only a few labeled cells were present after 15 days (Figure 3C). Virtually no labelled cells were seen at 0, 30, and 90 days (Figure 3D,E). In the center of the VCMX, MAC387^+^ cells appeared with a delay. Labeled cells were seen at 4 days (Figure 3F) and peaked at 7 days (Figure 3G), after which the number of labeled cells sharply declined (Figure 3H,I,K).

### 3.3. Immunohistochemistry with the Anti-PCNA Antibody

At 4 days, PCNA^+^ cells were seen in large numbers in the soft connective tissue next to the VCMX and also in some cells invading the peripheral portion of the VCMX (Figure 4A). The number of positive cells increased up to 15 days (Figure 4B,C) followed by a slow decline over time (Figure 4D,E). Up to 7 days, there were very few PCNA^+^ cells in the center of the VCMX (Figure 4F,G). The highest number of positive cells in the center was observed at 15 days (Figure 4H) followed by a slow decline up to 90 days (Figure 4I,K).

### 3.4. Quantitative Histological Analysis for MAC387 and PCNA

The counts of cells positive for MAC387 and PCNA are illustrated in Figure 5A,B, respectively. The number of MAC387^+^ cells reached its maximum at day 4 in the peripheral region of the biomaterial. At day 7, the number of MAC387^+^ cells in the periphery of the biomaterial was reduced to 50%, and at days 30 and 90 no MAC387^+^ cells could be detected anymore. In the central part of the VCMX, MAC387^+^ cells peaked at day 7, after which a sharp decline was observed. The highest number of MAC387^+^ cells was observed in the central part of the biomaterial.

In the peripheral part of the biomaterial, the number of PCNA^+^ cells increased from 4 to 15 days, followed by a decline. In the central part of the biomaterial, the first PCNA^+^ cells were detected at day 7, reached a maximum at 15 days, and declined thereafter. In the peripheral and central part of the biomaterial, the numbers of proliferative cells decreased from 15 days to 30 days by 25%, indicating that the proliferative phase was slowing down. After 90 days, the number of PCNA^+^ cells dropped by 75%. The highest number of PCNA^+^ cells was observed in the central part of the biomaterial.

### 3.5. Immunohistochemistry with the Anti-CD86 Antibody

In the marginal region, very weak labeling for CD86 was observed at 4 days in association with small blood vessels with thickened endothelial cells and in a few single scattered cells (Figure 6A). At 7 and 15 days, labeling of small blood vessels persisted and more scattered cells were labeled (Figure 6B,C). At 30 days, some blood vessels were still weakly positive, and the number of scattered positive cells had declined (Figure 6D). At 90 days, a few blood vessels and declined number of scattered cells were weakly positive (not shown). In the central region of the VCMX, no (Figure 6E) or very few (Figure 6F) faintly CD86^+^ scattered cells were seen at 4 and 7 days, respectively. Up to 7 days (Figure 6F), no positive staining of blood vessels was observed in the central part of the VCMX. At 15 and 30 days, the blood vessels in the center of the VCMX were medium-sized, their endothelial cells were flat, and immunolabeling was present (Figure 6G,H). Only very few scattered cells showed positive labeling (Figure 6G,H). Large multinucleated CD86^+^ cells were not observed at any healing time within the central pores of the VCMX. Only at 7 and 15 days did a few multinucleated giant cells showed a very weak labeling for CD86 at the periphery of the VCMX.

### 3.6. Immunohistochemistry with the Anti-TGM2 Antibody 

Numerous small blood vessels were seen in the soft connective tissue adjacent to the VCMX and started to invade the VCMX pores from day 4 onwards and reached the central pores at 15 days. Thus, blood vessel invasion was delayed in more central regions. Figure 7 shows immunostaining of blood vessels (asterisks) at day 7 at the periphery of the biomaterial (Figure 7A) but no positive reaction in the central region (Figure 7C). After 90 days, TGM2^+^ blood vessels were present in the soft connective tissue surrounding the residual VCMX matrix packages (Figure 7B,D).

## 4. Discussion

The soft tissue plays an important role in the maintenance of health around teeth and dental implants. Lack of an adequate volume of soft tissue can be corrected using autogenous soft tissue grafts or biomaterials. The advantage of using biomaterials is that no second surgical intervention is needed and that they are available at high quantities. The most widely used biomaterial for soft tissue augmentation is collagen-based, since it is believed that such a material mimics best the natural cell environment of the extracellular matrix [33], although other biomaterials are also candidates for soft tissue regeneration [28,34]. Collagenous matrices vary with respect to composition, three-dimensional structure, elasticity, and mechanical stability. If a collagen matrix collapses after implantation, host cell migration and blood vessel penetration may be impaired, which in turn negatively influences tissue degradation and integration as well as extracellular matrix production in the interior of the biomaterial. Thus, volume stability is an important parameter of biomaterials used for tissue augmentation. A second important factor is the inflammatory response to the implanted biomaterial. In the present study, we investigated the inflammatory response and cell migration into a novel volume-stable collagen matrix (VCMX), developed with the purpose of increasing the width of soft connective tissue.

The VCMX tested was described in three studies in canines where the focus was on volume gain and volume changes over time [21,35,36]. These studies have shown that gain of thickness of newly formed tissues adjacent to bone and dental implants was observed up to three months, but not so in the long-term, i.e., at six months. The authors concluded that the applied histomorphometrical techniques might not have been ideal to assess linear and volumetric tissue changes. However, the descriptive histology demonstrated favorable tissue integration in all three studies, but in none of the studies were the early healing events and the sequence of healing evaluated. As shown in the present study, the VCMX tested consists of an amorphous sheet-like matrix and small fibers and formed a trabecular structure containing large honeycomb-like pores. A recent study concluded that the sheet-like matrix was collagen and the small fibers were elastin [19]. In the present study, the histology showed that the VCMX did take up blood after its implantation but did not collapse (see Figure 1A,B). Other biomaterials used for soft tissue augmentation may collapse when they come in contact with liquids like blood. Recent studies have shown that invasion of host cells into a biomaterial is dependent on a pathway of interconnected pores [31,32]. Thus, interconnectivity of pores in the biomaterial scaffold and resistance to collapse are important requirements for tissue integration and regeneration.

The inflammatory and immune response determines the fate of a biomaterial and the type of tissue formed. In the worst case, a foreign body reaction may develop [33,37]. In the best scenario, wound healing is uneventful, displaying a short transient inflammatory phase after which the biomaterial becomes fully integrated in the host tissue. In this regard, cells from the monocyte/macrophage lineage play a pivotal role. Macrophages are among the cells that first come in contact with an implanted biomaterial. Macrophages possess a high plasticity; they are capable of polarizing from contributors of tissue inflammation (pro-inflammatory M1 macrophage) towards contributors of wound healing and tissue regeneration (anti-inflammatory M2 macrophage) [38]. M1 macrophages are an absolute requirement in the early wound healing phase. However, if the M1 macrophage phenotype persists and no shift to M2 macrophages occurs, an extended inflammatory phase ensues, eventually leading to the formation of multinucleated giant cells. Indeed, the prolonged presence of inflammatory cells is related to a foreign body reaction to the implanted biomaterial, especially during the first two to four weeks after implantation [39]. The present study has shown that MAC387^+^ (for macrophages) and CD86^+^ (for both multinucleated giant cells and M1 macrophages) [24] cells are only present for a short period of time. Furthermore, the few CD86^+^ multinucleated giant cells were only detected at days 7 and 15 and were exclusively at the peripheral interface between the biomaterial and surrounding soft connective tissue. These cells may have a function in the degradation of the biomaterial. CD86 is traditionally used as an M1 macrophage marker, but it is reported that endothelial cells can also express CD86 [40]. In the present study, CD86^+^ newly formed blood vessels were present at the marginal portion of the biomaterial and approaching the biomaterial center in less than a week.

Apart from being key players during the inflammatory phase, macrophages have an influence on angiogenesis [41,42] and fibroblast proliferation and activity [43]. In the present study, the first blood vessels invading the biomaterial were seen after 4 days and reached the center at day 15. Another factor contributing to invasion and proliferation of cells is biomaterial micro-architecture and composition [44]. For example, a biomaterial with interconnected pores supports invasion and ingrowth of cells [45]. This not only helps ingrowth of new cells but also facilitates nutrient and oxygen diffusion [45]. In addition, a certain pore size and interconnectivity of pores facilitate angiogenesis [46]. The favorable three-dimensional structure of the VMCX may have contributed to the observed fast ingrowth of blood vessels. Whether or not macrophages contributed to angiogenesis in our experimental setting requires further studies. However, what is known is that anti-inflammatory M2 macrophages, rather than pro-inflammatory M1 macrophages, promote angiogenesis [41]. Unfortunately, we could not find antibodies against M1 and M2 macrophages that worked in canine tissues. We tested about six other anti-macrophage antibodies without having success, and the anti-MAC387 antibody was the only anti-macrophages antibody that was working in canine tissues. Regarding cell proliferation, we have shown that the pores of the VCMX were not only rapidly invaded by blood vessels but also by other fast proliferating cells. The proliferative PCNA^+^ cells started to invade the VCMX after 4 days and reached the center after 7 days. The peak of the proliferative phase was at day 15 with a short delay in the central part of the biomaterial. Since no inflammatory cells were detected after 15 days, it seems likely that these cells were mesenchymal cells, probably fibroblasts. The morphology of these cells supports this assumption. Fast ingrowth of blood vessels coupled with fast proliferating cells such as fibroblasts, which eventually will start to produce collagen as part of the newly formed extracellular matrix, ensures rapid integration of the biomaterial into the host tissue. This, in turn, may assist in maintaining the volume of the implanted biomaterial, as part of it may undergo degradation over time.

The present study provides sound biological knowledge about the tissue response towards a new biomaterial for soft tissue augmentation. Nevertheless, the study has limitations. One of the limitations of the present study is the small sample size. We therefore consider this experiment as a pilot study to explore the possibilities using immunohistochemical techniques to evaluate the tissue response towards a biomaterial in the canine, an animal very rarely used for biomaterial testing. In this sense, the study is unique and provides very valuable data for future studies in the biomaterial field. The results are very coherent, showing a logical sequence of events with a short inflammatory phase and a rapid ingrowth of non-inflammatory cells. Further studies may use a higher number of samples and may also address the tissue response and integration of this biomaterial in humans.

## 5. Conclusions

The VCMX appeared volume-stable and elicited a short inflammatory phase and fast ingrowth of blood vessels and non-inflammatory cells resulting in a complete integration in the host tissue. Overall, the present study demonstrates a high biocompatibility of the VCMX, which makes it a potent candidate for soft connective tissue augmentation in dentistry and possibly also other disciplines.

## Figures and Tables

**Figure 1 materials-12-03721-f001:**
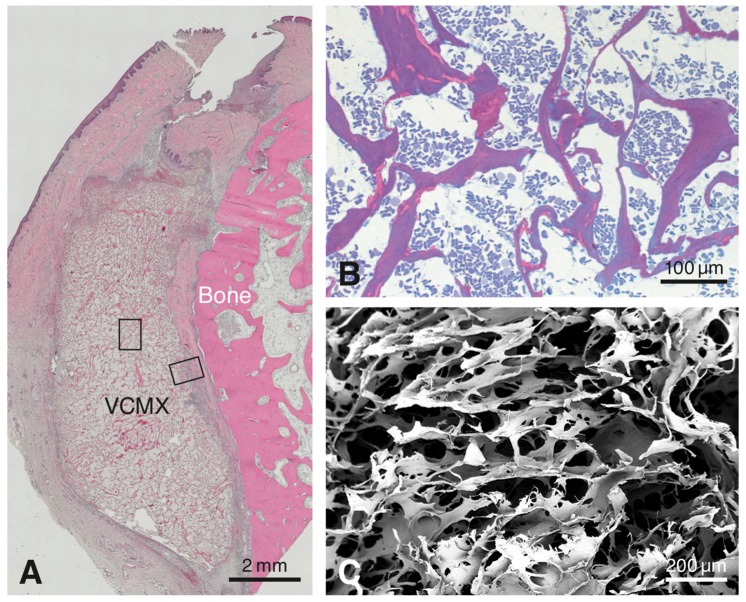
(**A**) Overview of a paraffin-embedded tissue section stained with Hematoxylin and Eosin showing the volume-stable collagen matrix (VCMX) at 4 days after implantation lying parallel to the bone. The right rectangle denotes the VCMX periphery facing bone, whereas the other rectangle marks the VCMX center. (**B**) The resin section illustrates the VCMX pores filled with blood plasma and mainly erythrocytes at 4 h (= day 0) after implantation. (**C**) The scanning electron microscopic image illustrates the three-dimensional-structure and pores of the VCMX.

**Figure 2 materials-12-03721-f002:**
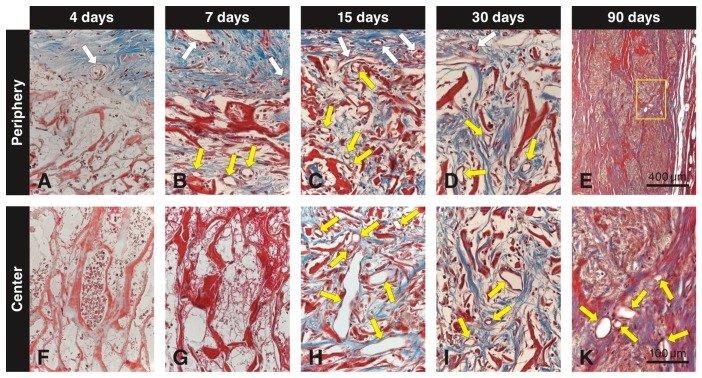
Paraffin-embedded tissue sections stained with Masson’s trichrome showing the VCMX at different time points and in two different regions (**A**–**I**,**K**). While white arrows point to blood vessels in the surrounding soft connective tissue, yellow arrows mark blood vessels in the pores of the VCMX. Blood vessels inside the VCMX become clearly visible from day 7 on at the periphery and from day 30 on in the center.

**Figure 3 materials-12-03721-f003:**
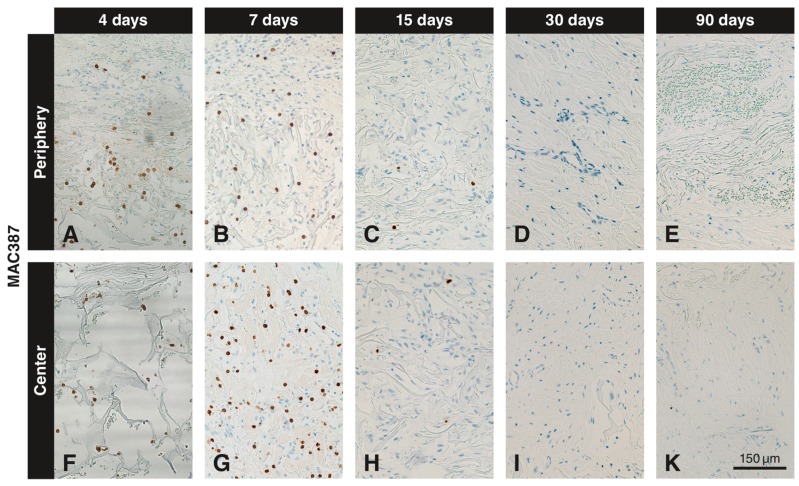
Inmunohistochemical staining with the anti-MAC387 antibody at different time points and in peripheral and central regions of the VCMX. Brown (DAB^+^) cells indicate presence of macrophages. Four days after implantation, MAC387^+^ cells are present in the soft connective tissue surrounding the VCMX and in the VCMX pores (**A**,**F**). Most MAC387^+^ cells are seen in the peripheral part at 4 days (**A**) and in the center at 7 days (**G**), while they decrease on the periphery (**B**). Thereafter, the number of positive cells sharply decreased (**C**,**H**). No labelled cells can be seen after 30 (**D**,**I**) and 90 days (**E**,**K**) of implantation.

**Figure 4 materials-12-03721-f004:**
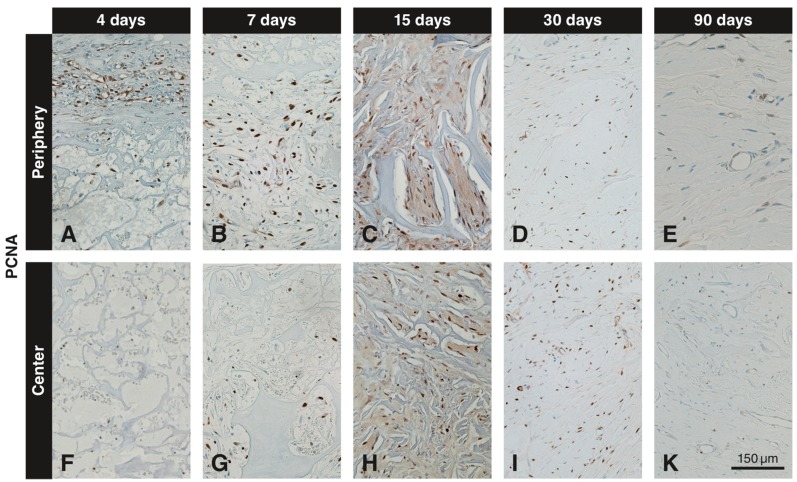
Inmunohistochemical staining for PCNA to detect proliferating cells at different time points in peripheral and central regions of the VCMX. Brown (DAB^+^) nuclei indicate proliferating cells. PCNA^+^ cells are present in large numbers outside and inside (periphery) of the VCMX at day 4 (**A**), whereas the VCMX center is devoid of labeled cells (**F**). While the number of PCNA^+^ cells increases in the periphery up to 15 days (**B**,**C**), the first PCNA^+^ cells in the center appear on day 7 (**G**) and increase on day 15 (**H**). The number of PCNA^+^ cells gradually decreased over time both in the periphery (**D**,**E**) and in the center (**I**,**K**) of the VCMX.

**Figure 5 materials-12-03721-f005:**
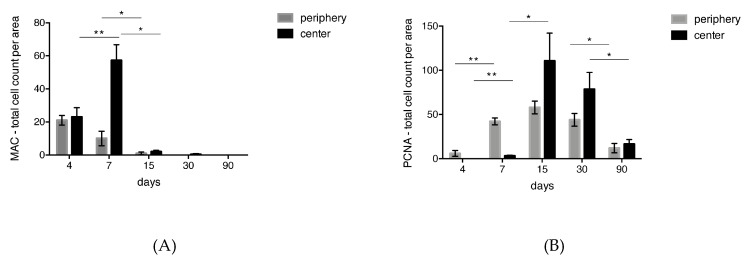
Bar charts showing the quantitative analysis (labeled cells/0.36 mm^2^) of (**A**) MAC387^+^ and (**B**) PCNA^+^ cells throughout the different time points. Cells positive for MAC387 peaked at the margins of the VCMX at day 4, while the peak at the center was at 7 days. PCNA^+^ cells peaked at day 15 and reduced gradually over time. * *p* < 0.05, ** *p* < 0.01.

**Figure 6 materials-12-03721-f006:**
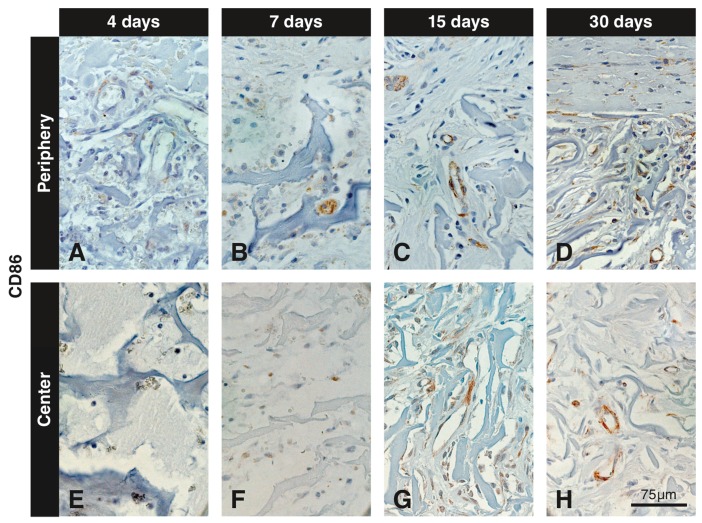
Inmunohistochemical staining for CD86 at different time points in peripheral and central regions of the VCMX. Brown (DAB^+^) staining indicates CD86^+^ macrophages and endothelial cells. At day 4, a few single scattered cells and blood vessels in the peripheral pores of the VCMX are labeled for CD86 (**A**), whereas the center does not show any labeled cells (**E**). At days 7 and 15, the number of CD86^+^ scattered cells and endothelial cells had increased at the periphery (**B**,**C**). In the central pores the first CD86^+^ cells appeared after 7 days (**F**) and increased at 15 days (**G**). After 30 days, the number of CD86^+^ scattered cells had decreased but blood vessels were still positive (**D**,**H**).

**Figure 7 materials-12-03721-f007:**
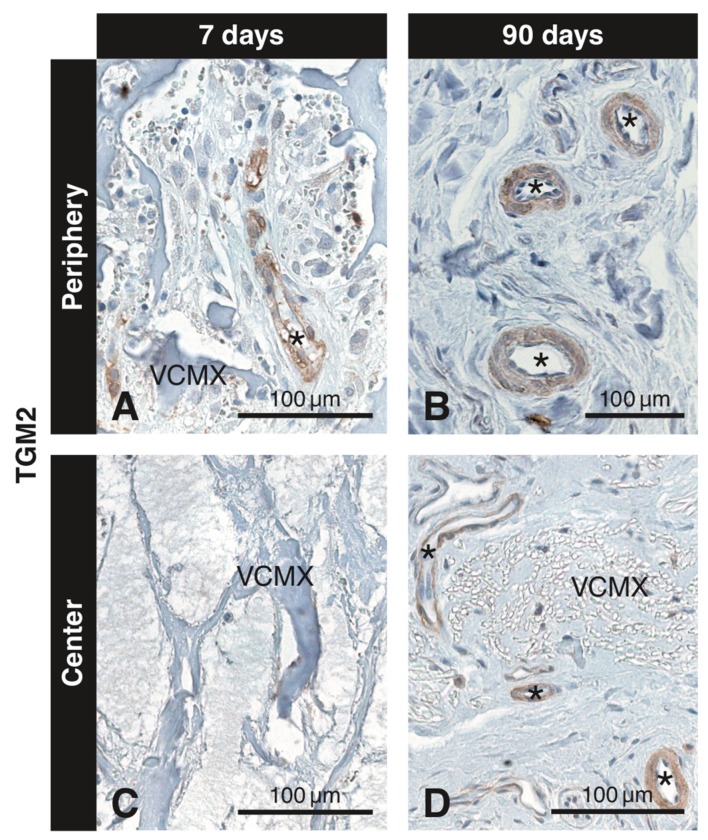
Inmunohistochemical staining for TGM2 to detect blood vessels at different time points in peripheral and central regions of the VCMX. Brown (DAB^+^) staining indicates TGM2^+^ endothelial cells. While the first TGM2^+^ blood vessels are seen at day 7 in the peripheral pores of the VCMX (**A**), the VCMX center is devoid of blood vessels at that time (**C**). At 90 days, TGM2^+^ blood vessels are present in the soft connective tissue intermingling with packages of residual VCMX (**B**,**D**).

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
