# Peer review of "Tissue Response to a Porous Collagen Matrix Used for Soft Tissue Augmentation"

_materials, 2019, doi:10.3390/ma12223721_

Round 1

Reviewer 1 Report

The authors demonstrated that volume-stable collagen matrix (VCMX) elicited a short inflammatory phase followed by rapid tissue integration. I think that this manuscript is valuable because it provides sound biological knowledge about the tissue response towards a new biomaterial for soft tissue augmentation. However, they should revise the following points.

As they noticed, the sample size is small. In order to strengthen their results, are there any related reports using canine or other animals. If so, they should discuss about it. They just examined the local inflammatory responses. As this study is preclinical one, they should describe the whole body reaction, especially the peripheral blood changes.

Author Response

Response to Reviewer 1. All changes are highlighted in red in the manuscript.

The authors demonstrated that volume-stable collagen matrix (VCMX) elicited a short inflammatory phase followed by rapid tissue integration. I think that this manuscript is valuable because it provides sound biological knowledge about the tissue response towards a new biomaterial for soft tissue augmentation. However, they should revise the following points.

Response: Thank you for your positive comments and your very valuable suggestions to improve the quality of our manuscript.

As they noticed, the sample size is small. In order to strengthen their results, are there any related reports using canine or other animals. If so, they should discuss about it.

Response: Yes, there are a few reports with identical or similar collagen scaffolds.

In Thoma et al. (J Clin Periodontol 2011;38:1063), the focus was on tissue thickness gain and maintenance. They showed that the collagen scaffold demonstrated favorable tissue integration and tissue gain comparable to the gold standard, the autogenous subepithelial connective tissue graft. In Thoma et al. (Clin Oral Implants Res 2012;23:1333), the tissue response of collagen scaffold with high or low levels of chemical cross-linking was evaluated in athymic mice. Since we investigated the inflammatory response and the athymic mice is not an appropriate model for this, we cannot compare our results with those from this study in mice. In Thoma et al. (Clin Oral Implants Res 2015, cross-linked and non-crosslinked collagen scaffolds with or without rhPDGF-BB were tested in a rat ectopic model. They concluded that the spongeous cross-linked collagen scaffold facilitated early vascularization and demonstrated network presence over a longer time span. In Thoma et al. (J Clin Periodontol 2017;44:185), soft tissue volume augmentation at dental implants were evaluated with the volume-stable three-dimensional collagen matrix. They demonstrated that soft tissue volume augmentation was maintained to a similar extent for the biomaterial and the autogenous subepithelial connective tissue graft for up to 2 months. Naenni et al. (Clin Oral Invest 2018;22:1185) showed that the augmentation volume decreased to a level close to the pre-surgery situation.

In conclusion, we could identify 3 studies in dogs. In all 3 studies, the focus was on volume gain and changes over time. All studies demonstrated favorable tissue integration. However, in none of the studies the early healing sequence was analyzed and no immunohistochemistry was applied for a more detailed analysis characterizing cell proliferation, inflammatory cells or blood vessels.

We have added the following text in the discussion on p.10, line 320: “The VCMX tested was described in three studies in canines where the focus was on volume gain and volume changes over time (Thoma et al. J Clin Periodontol 2011;38:1063, Thoma et al. J Clin Periodontol 2017;44:185, Clin Oral Invest 2018;22:1185). These studies have shown that gain of thickness of newly formed tissues adjacent to bone and dental implants was observed up to 3 months, but not so in the long-term, i.e. at 6 months. The authors concluded that the applied histomorphometrical techniques might not have been ideal to assess linear and volumetric tissue changes. However, the descriptive histology demonstrated favorable tissue integration in all three studies, but in none of the studies the early healing events and the sequence of healing were evaluated.”

They just examined the local inflammatory responses. As this study is preclinical one, they should describe the whole body reaction, especially the peripheral blood changes.

Response: We indeed concentrated on the events inside the biomaterial, particularly on the dynamic events during repopulation of the biomaterial. We added now a paragraph on the wound healing of the tissues surrounding the biomaterial on p.4, line 180: “The VCMX was clearly identifiable and demarcated from the surrounding tissues up to 30 days. At 4 days (Fig. 2A), the connective tissue surrounding the VCMX showed the typical feature of granulation tissue, i.e. presence of a fibrin network, many erythrocytes, small blood vessels, and some leukocytes. Initial invasion of blood vessels, leukocytes and mesenchymal cells into the VCMX pores was observed at 4 days and restricted to the border region only. At 7 days (Fig. 2B), larger blood vessels and many mesenchymal cells were seen surrounding and invading the collagen scaffold. At 15 days (Fig. 2C), still many blood vessels, few inflammatory cells, and many spindle-shaped fibroblasts oriented parallel to newly formed collagen fibers were seen external to the biomaterial. Some residual biomaterial became integrated in the surrounding soft connective tissue. At 30 days (Fig. 2D), the surrounding host collagen matrix appeared more mature, the epithelial layer was closed and well-keratinized. At the border, the biomaterial was well-integrated in the surrounding more mature soft connective tissue and inflammatory cells were scarce. At 90 days (Fig. 2E) , residual islands of biomaterial were dispersed and well-integrated in the surrounding host soft connective tissue.”

Reviewer 2 Report

Minor comments:

Statistical analysis should be provided in Figure 5. The use of animals in biomedical research have to be implemented in accordance to the Directive 2010/63/EU on the protection of animals used for scientific purposes and the authors should get a permission from bioethical committee. This should be indicated in the section 2.1.

Author Response

Response to Reviewer 2. All changes are highlighted in red in the manuscript.

Statistical analysis should be provided in Figure 5. The use of animals in biomedical research have to be implemented in accordance to the Directive 2010/63/EU on the protection of animals used for scientific purposes and the authors should get a permission from bioethical committee. This should be indicated in the section 2.1.

Response: Thank you for your valuable comment. We have now added the statistics in Fig. 5 on p.8 and in the Materials & Methods section on p.4.

New text on p.4: “The paired t-test was used and the statistical significance was established at P<0.05.”

New text on in Fig.5 p.8: “* P < 0.05, ** P < 0.01.”

We also have added on p.3, line 96: “This preclinical study protocol has been approved by the Ethical Committee of NAMSA”.

Reviewer 3 Report

Comments

Overall the article “Tissue response to a porous collagen matrix used for soft tissue  augmentation” is interesting and present good-looking data on the in vivo use of highly porous and volume-stable collagen matrix (VCMX), for  soft dental, soft tissue regeneration. The idea of using porous scaffolds to facilitate cell proliferation and of using the foreign-body response, to generate ECM scaffolds are not novel but the combination of the two retain a certain level of novelty. Some minor comments particularly literature cited and figure legends needs to be worked out before accepting it for the publications.

In general, the introduction is too general and specific influence of designed ECM for the tissues of interest should be taken into account and discussed here.

Very little information is provided about the porosity, swelling and mechanical strength and controlled release of incorporated biological fluid for homogenous tissue augmentation.

In introduction, authors overlooked some of recent reports in context with proliferation of cells into the biomaterial as crucial step for the integration of the biomaterial in the host tissue, denoting a high biocompatibility as explained in context with referred work as citation [28]. Few very recent reports with novel strategy to promote scaffold integrity with surrounding tissue biomaterial has been shown in PMID: 30959985 & 31382208. Please cite these reports.

I could not observe any orange arrow labelling in figure 2? Does authors mean white arrow? Please edits figure legends since it quite confusing as hff3ff

Authors are suggested to provide more details in figure legends with color coded structures in the figures. For example in figure 3 A,B,G,F what are brown dots, do they represents some specific mineralized cells?

Did authors observed did any statistical analysis on quantitative data for MAC387 and PCNA positive cells at day 4 and 7 shown in figure 5? Authors are requested to provide standard deviation of uncertainty or a particular confidence interval (e.g., a 95% interval) observed in bar chart in figure 5.

Sentence “The quantitative analysis of the PCNA labeling is shown in figure 5B.” is part of section 3.4. Quantitative histological analysis for MAC387 and PCNA which is mentioned in section 3.3.

A lot of research articles and reviews are available on biomaterial for soft tissue augmentation in context with collagen-based scaffolds and their importance in material mimesis in the natural cell environment of the extracellular matrix as discussed in this work in discussion section. Only one has been cited in this work as reference 31, so it is not clear if the authors has done proper analysis of the previous research before making any conclusions in this work. Please cite a review explaining these concept in fine details published with DOI https://doi.org/10.4081/vl.2018.7196.

Author Response

Response to Reviewer 3. All changes are highlighted in red in the manuscript.

Overall the article “Tissue response to a porous collagen matrix used for soft tissue  augmentation” is interesting and present good-looking data on the in vivo use of highly porous and volume-stable collagen matrix (VCMX), for  soft dental, soft tissue regeneration. The idea of using porous scaffolds to facilitate cell proliferation and of using the foreign-body response, to generate ECM scaffolds are not novel but the combination of the two retain a certain level of novelty. Some minor comments particularly literature cited and figure legends needs to be worked out before accepting it for the publications.

Response: Thank you for your positive comments and your very valuable suggestions to improve the quality of our manuscript.

In general, the introduction is too general and specific influence of designed ECM for the tissues of interest should be taken into account and discussed here.

Response: We have added the following text in the introduction on p.2, line 61:Such a biomaterial must be biocompatible, allow ingrowth of blood vessels and progenitor cells, and withstand mechanical forces generated as a result of suturing, wound contraction, and mastication, thereby maintaining tissue volume.”

Very little information is provided about the porosity, swelling and mechanical strength and controlled release of incorporated biological fluid for homogenous tissue augmentation.

Response: We have added information on the VCMX from the literature. New text on p.2, line 63: “The collagen matrix consists of 60-96% (w/w) porcine collagen type I and III and 4-40% (w/w) elastin, has an average pore diameter of 92 µm and 93% volume porosity with interconnected pores. Scaffold stiffness was achieved by chemical cross-linking. The scaffold remained elastic even after mechanical forces were applied for 14 days, as evaluated in a dynamic bioreactor test system mimicking the mechanical stresses of the human in vivo situation (Mathes et al. Biotechnology and Bioengineering 2010;107:1029-1039).”

In introduction, authors overlooked some of recent reports in context with proliferation of cells into the biomaterial as crucial step for the integration of the biomaterial in the host tissue, denoting a high biocompatibility as explained in context with referred work as citation [28]. Few very recent reports with novel strategy to promote scaffold integrity with surrounding tissue biomaterial has been shown in PMID: 30959985 & 31382208. Please cite these reports.

Response: We have added these two papers in the introduction on p.2, line 79. New text: “Proliferation of cells into the biomaterial is crucial for the integration of the biomaterial in the host tissue, denoting a high biocompatibility (28, Tiwari et al. Polymers 2019;11,1;doi:10.3390/polym11010001 = PMID: 30959985). Furthermore, the rigidity of a biomaterial determines cell invasion and the cell type into which mesenchymal cells will differentiate (Singh et a. Biomaterials 2019 PMID:31382208)”.

I could not observe any orange arrow labelling in figure 2? Does authors mean white arrow? Please edits figure legends since it quite confusing as hff3ff

Response: You are right, the arrows should be white. New text in legend of Fig. 2. line 262 on p.7: “While white arrows point to blood vessels in the ….”

Authors are suggested to provide more details in figure legends with color coded structures in the figures. For example in figure 3 A,B,G,F what are brown dots, do they represents some specific mineralized cells?

Response. In Fig. 3, the brown dots are DAB-labeled cells. An anti-macrophage antibody was used here to label macrophages. New text in legend of Fig. 3 on p.7: “Brown (DAB+) cells indicate presence of macrophages.”

In Fig. 4, the brown dots are DAB-labeled nuclei. An anti-PCNA antibody was used here to label cells in the proliferative phase. New text in legend of Fig. 4 on p.8: “Brown (DAB+) nuclei indicate proliferating cells.”

In Fig. 6,An anti-CD86 antibody was used here to label cells of the macrophage lineage expressing an M1 phenotype as well as endothelial cells. New text in legend of Fig. 6 on p.9: “Brown (DAB+) staining indicates CD86+macrophages and endothelial cells.”

In Fig. 7, An anti-TGM2 antibody was used here to label endothelial cells. New text in legend of Fig. 7 on p.10: “Brown (DAB+) staining indicates TGM2+endothelial cells.”

Did authors observed did any statistical analysis on quantitative data for MAC387 and PCNA positive cells at day 4 and 7 shown in figure 5? Authors are requested to provide standard deviation of uncertainty or a particular confidence interval (e.g., a 95% interval) observed in bar chart in figure 5.

Response: Thank you for your valuable comment. We have now added the statistics in Fig. 5 on p.8 and in the Materials & Methods section on p.4.

New text on p.4: “The paired t-test was used and the statistical significance was established at P<0.05.”

New text on in Fig.5 p.8: “* < 0.05, ** < 0.01.”

Sentence “The quantitative analysis of the PCNA labeling is shown in figure 5B.” is part of section 3.4. Quantitative histological analysis for MAC387 and PCNA which is mentioned in section 3.3.

Response: Thank you. We have removed “The quantitative analysis of the PCNA labeling is shown in figure 5B.” and “The quantitative analysis of the MAC387 labeling is shown in figure 5B.”

A lot of research articles and reviews are available on biomaterial for soft tissue augmentation in context with collagen-based scaffolds and their importance in material mimesis in the natural cell environment of the extracellular matrix as discussed in this work in discussion section. Only one has been cited in this work as reference 31, so it is not clear if the authors has done proper analysis of the previous research before making any conclusions in this work. Please cite a review explaining these concept in fine details published with DOIhttps://doi.org/10.4081/vl.2018.7196.

Response: The two reviews we have chosen, i.e. No. 28 and No. 31, cover the issue of extracellular matrix-based biomaterial scaffolds very well, but we are very open to add another reference like the one suggested by reviewer No. 3. Thus, we have changed the text in the discussion on p.10, line 308-310 as follows: “The most widely used biomaterial for soft tissue augmentation is collagen-based, since it is believed that such a material mimics best the natural cell environment of the extracellular matrix (31), although other biomaterials are also candidates for soft tissue regeneration (28, Singh et al. Veins and Lymphatics 2018;7:7196).